# Mental Health Difficulties and Countermeasures during the Coronavirus Disease Pandemic in Japan: A Nationwide Questionnaire Survey of Mental Health and Psychiatric Institutions

**DOI:** 10.3390/ijerph18147318

**Published:** 2021-07-08

**Authors:** Tomohiro Nakao, Keitaro Murayama, Sho Takahashi, Mami Kayama, Daisuke Nishi, Toru Horinouchi, Nozomu Oya, Hironori Kuga

**Affiliations:** 1Department of Neuropsychiatry, Graduate School of Medical Sciences, Kyushu University, Fukuoka 8128582, Japan; 2Department of Psychiatry, Kyushu University Hospital, Fukuoka 8128582, Japan; murayama.keitaro.003@m.kyushu-u.ac.jp; 3Department of Disaster and Community Psychiatry, Faculty of Medicine, University of Tsukuba, Tsukuba 3058577, Japan; shotaka72@gmail.com; 4Psychiatric & Mental Health Nursing, Graduate School of Nursing, St. Luke’s International University, Tokyo 1040044, Japan; mkayama@slcn.ac.jp; 5Department of Mental Health, Graduate School of Medicine, The University of Tokyo, Tokyo 1130033, Japan; d-nishi@m.u-tokyo.ac.jp; 6Department of Psychiatry, Hokkaido University Graduate School of Medicine, Sapporo 0600808, Japan; tetsukawa1234@gmail.com; 7Department of Psychiatry, Graduate School of Medical Science, Kyoto Prefectural University of Medicine, Kyoto 6028566, Japan; n-oya@koto.kpu-m.ac.jp; 8National Center for Cognitive Behavior Therapy and Research, National Center of Neurology and Psychiatry, Tokyo 1878551, Japan; hirokuga@ncnp.go.jp

**Keywords:** COVID-19, mental health, questionnaire survey, mental health and welfare centers, psychological intervention

## Abstract

The number of people with coronavirus disease (COVID-19) has been increasing worldwide. Anxiety about potential infection, fear of severe illness, death, economic problems, and loneliness and isolation brought on by social distancing are increasingly being experienced by people. Therefore, it is imperative to address and improve such mental health-related problems during COVID-19. We aimed to investigate the current mental health care and psychological intervention statuses related to COVID-19 in Japan. In a questionnaire survey, 55 of 69 (80%) mental health and welfare centers and 194 of 931 (21%) psychiatric institutions across Japan responded. COVID-19 patients, their family members, and the general public often consulted the mental health and welfare institutions through telephone. The questionnaire included various information of mental health difficulties related to COVID-19 such as the numbers and contents of the consultations, and the type of the interventions. The contents of consultation included psychological symptoms (anxiety, depression, insomnia, and alcohol problems) and psychosocial problems (interpersonal problems, prejudice, and discrimination). Overall, 9% of mental health and welfare centers provided psychological first aid as psychological intervention and 28% of consultations involved cases requiring urgent care. In Japan, consultations about COVID-19-related mental health problems occurred mainly in mental health and welfare centers. There is urgent need to establish a system that enables mental health triage and brief psychological interventions that are feasible in the centers.

## 1. Introduction

A novel coronavirus disease (COVID-19) that causes severe acute respiratory syndrome was identified in December 2019, and the disease has rapidly spread worldwide. The COVID-19 pandemic has persisted through 2021 and caused extensive damage globally. As of February 2021, the total number of infected people worldwide exceeded 100 million, and the total number of deaths exceeded 2.3 million [1]. In Japan, as of December 2020, the number of infected people has been rapidly increasing nationwide [2] (Figure 1). Among the general population, there has been an ongoing experience of anxiety about potential infection, fear of severe illness, death, and related economic problems as well as of loneliness and isolation due to social distancing measures. Under these circumstances, it is essential to maintain and improve the resilience of people and mental health.

Worldwide, there has been an increase of mental health difficulties during the COVID-19 pandemic [3]. Incidences of psychosis, mood disorders, and anxiety disorders are significantly higher in people following COVID-19 diagnosis than the incidences in people with other infections such as influenza [4]. In addition, COVID-19 is having a major impact on mental health in infected and uninfected people. In the United States, approximately 45% of the population has experienced psychological distress during the COVID-19 pandemic [5]. A longitudinal study of psychiatric diseases in the United States demonstrated that history of psychiatric disease and reduced income due to the pandemic were associated with increased risk of COVID-19 [6]. A Chinese study of the impact of COVID-19 on the mental health of 1434 patients with ill mental health (unknown diagnoses) reported that 300 (20.9%) experienced worsening of mental symptoms during the COVID-19 pandemic [7]. Health care workers also experienced increases in symptoms of depression, anxiety, and obsessive-compulsive symptom, whereas the general population experienced increases in anxiety and depression and a decreased feeling of well-being. The impact of COVID-19 on mental health is serious, and the United Nations and the World Health Organization (WHO) [8] have urged countries to strengthen their countermeasures.

In Japan, the number of mental health consultations related to COVID-19 in mental health administrative services increased approximately 2.8 times from April to May 2020 [9]. However, the related specific measures and early intervention methods are not fully developed, and there are no standard guidelines or protocols. Furthermore, in Japan, there are no appropriate countermeasures against the increased depression and suicide rates. The suicide rate in Japan is higher than that in other developed countries. Ten years ago, the annual number of suicides was approximately 30,000 (22.6 per 100,000 population), and with increased national efforts in the last 10 years, the annual number of suicides decreased to approximately 20,000 (15.7 per 100,000 population) (Figure 2) [10]. However, a research group at Kyoto University has suggested that the annual number of suicides in Japan may increase to 40,000 due to the influence of COVID-19 [11]; therefore, suicide prevention is an urgent issue.

In Japan, institutions called “mental health and welfare centers” provide public mental health services. Mental health and welfare centers are in 47 prefectures and 22 government-designated cities to disseminate knowledge, conduct research, and provide consultations and guidance regarding the community mental health and welfare of patients with a mental health disorder. They also play a central role in improving the community’s mental health and well-being. The staff consists of psychiatrists, nurses, psychologists, and psychiatric social workers. All residents in Japan can receive consultations for mental health difficulties free of charge. However, mental health and welfare centers do not provide pharmacotherapy or intensive psychotherapy since they are not medical institutions. Therefore, individuals requiring such care are referred to psychiatric institutions.

In Japan, mental health and welfare centers are providing consultations for COVID-19-related mental health difficulties ranging from mild to severe, indicating that the centers are playing a key role in mental health services. However, to our knowledge, there are no data regarding the number of monthly consultations, contents of consultation, psychological intervention performed, or the current status of remote consultations in mental health and welfare centers. Therefore, this study aimed to determine the current status of mental health care (psychological assessment and intervention methods) related to COVID-19, provided by mental health and psychiatric care workers in mental health and welfare centers across the country, using a questionnaire survey.

## 2. Materials and Methods

### 2.1. Study Design

A questionnaire survey was conducted to collect information from study institutions from 1 November to 31 December 2020.

### 2.2. Study Institutions

This study included 1000 institutions, consisting of all mental health and welfare centers (n = 69) and psychiatric institutions (n = 931) selected from 1200 specialized psychiatric hospitals, 1600 psychiatric clinics, and 890 psychiatric departments of university and general hospitals in Japan, via stratified random sampling. The institutions were assigned a randomly generated number in an Excel spreadsheet, and the list of random numbers was then sorted in the ascending order. From this list, we sequentially selected the institutions.

### 2.3. Questionnaire Survey

Request letters (providing an explanation on the study) and questionnaires were mailed by to the head of institutions included in the study. Responses to the questionnaire were made based on the consultation records and medical records of staff after obtaining consent from the head of institution. Returned questionnaires (paper or web-based versions) by staff of institutions to the research company were regarded as consent to participate in this study. The following information was obtained from the returned questionnaires (obtained information). Details are shown in the attached questionnaire form.

#### 2.3.1. Basic Information

The obtained information from the questionnaire was as follows: type of study institution, location by region, covering population size, the number of consultations on psychological problems (including sleep disorder) related to COVID-19, the number of monthly consultations from 1 January to 31 October 2020, the number of consultations by sex and age group, and category of consulter (e.g., non-infected individuals, infected individuals, or family members of infected individuals).

#### 2.3.2. Information on Contents of Consultation

The contents of consultation were categorized according to whether the mental health difficulties were new or pre-existing; the contents of the mental health difficulty; and the presence or absence of consultation requiring urgent care.

#### 2.3.3. Information on Psychological Intervention

Data on the presence or absence of psychological intervention (F code in the International Statistical Classification of Diseases and Related Health Problems 10th Revision [ICD-10]) classification, symptoms requiring psychological intervention, the number and contents of psychological interventions, and the number of sessions and intervention duration were collected.

#### 2.3.4. Remote Consultation

Information was obtained via remote consultation as follows: category of consulter (e.g., patients, family members, and local residents), category of consultation (e.g., the infection itself, discrimination related to infection, interpersonal problems), reception hours for telephone consultation, how to access consultation service (e.g., through homepage, public relations magazines, and posters), whether or not the time limit for responding to email consultation was fixed, the department in charge of consultation, the number of consultation staff, how to make consultation service well known, the presence or absence of support for consultation staff, the number of monthly consultations, contents of consultation, and difficulties with remote consultation (optional comment).

#### 2.3.5. Data Analysis

The data obtained were anonymized, and statistical analysis was performed by INTAGE Research Inc. Simple tabulation and cross-tabulation were performed using the Lyche-Epoch aggregation tool. The anonymized data, correspondence table, and aggregate results were stored by Kyushu University.

## 3. Results

### 3.1. Basic Information

Responses were obtained from a total 249 of 1000 institutions surveyed (response rate, 25%), consisting of 55 mental health and welfare centers (response rate, 80%), 67 university or general hospitals (response rate, 30%), 84 psychiatric clinics (response rate, 21%), and 43 psychiatric specialized hospitals (response rate, 14%). Data obtained from the 55 mental health centers (MH) and the 194 psychiatric units (PU) were compared.

The number of institutions that were consulted about mental health related to COVID-19 was 53 (96%) and 84 (43%) in the MH and PU group, respectively, indicating that mental health administrative services play a central role in mental health consultations in Japan. In the MH group, telephone consultations were conducted in all 53 (100%) institutions and face-to-face consultations or examinations in 15 (28%). The mean number of telephone consultations per institution from January to October, 2020 was 236 (range: 17–993), whereas the mean number of consultations per institution was 11 (range: 0–87). Thus, telephone consultation was the most frequently used method in the MH group. In the PU group, telephone consultations were conducted in 22 (26%) institutions, while face-to-face consultations or examinations were conducted in most of the institutions (80 institutions, 95%), indicating that face-to-face examinations were provided for most individuals who consulted in PUs, although the number of consultations was small (Table 1).

There were no consultations in any of the institutions in January 2020, when COVID-19 started to spread. In the MH group, the number of monthly consultations gradually increased between February and March 2020 and exceeded 50 per month in April and May, during which the state of emergency declaration was made due to the rapid increase in the number of infected people. Thereafter, the number of monthly consultations decreased with the number of infections to 18 per month in October (Figure 3). In the PU group, no marked increase or decrease was found, and the mean number of monthly consultations per institution was around one.

Regarding the differences by age group, the number of consultations was high among individuals in their 40s and 50s (Figure 4).

Regarding the differences by sex, the number of consultations by females was more than those of males in the MH (152 vs. 87) and the PU group (6 vs. 3).

The mean number of consultations according to the categories of consulters in the MH group was 270 for non-infected individuals, 11 for infected individuals, 1.1 for family members of infected individuals, 0.9 for suspected infected individuals, and 3.7 for health care workers. Thus, most consultations were made by non-infected individuals, and a similar tendency was observed in the PU group.

### 3.2. Information on Contents of Consultation

The contents of consultation in the MH group varied, including on the infection itself, discrimination related to infection, interpersonal problems, work and economic problems, and mental difficulties. On the other hand, the contents of consultation in the PU group were mostly about mental problems. Approximately 80–90% of institutions were consulted about mental difficulties in both groups (Figure 5). Of 52 and 72 institutions in the MH and PU groups that were consulted about mental difficulties, 49 (94%) and 28 (39%) were for both new and pre-existing mental difficulties (MH group) and new mental problems (PU group), respectively.

The mean number of consultations per institution for the most frequent mental difficulties in the MH and PU groups was reported for anxiety (84 vs. 5), depressive mood and loss of motivation (24 vs. 3.9), interpersonal and child problems (20 vs. 0.7), psychosomatic problems (19 vs. 3.2), insomnia (13 vs. 3.8), frustration and irritability (12 vs. 1.7), and increased amount of drinking (10 vs. 0.3), respectively. There were also consultations for suicidal ideation (6.0 vs. 0.4) and suicidal attempt (0.3 vs. 0.1), respectively, although the numbers were small (Figure 6).

Consultations requiring urgent care were reported in 15 (28%) and 28 (33%) MH and PU institutions, with a mean number of consultations requiring urgent care of 5.6 and 4.3, respectively.

### 3.3. Information on Psychological Intervention

Psychological intervention was performed in 22 (41%) and 61 (72%) institutions that were consulted, in the MH and PU groups, respectively. The symptoms requiring psychological intervention were insomnia, depression, abuse-related problems, and posttraumatic disorder in the MH group. The contents of psychological intervention in the MH group included health consultation (59% of 22 institutions that provided psychological intervention), utilization of brochures (53%), holding of workshops (32%), individual counseling (40%), group counseling (13%), and referral to medical institutions (63%). Psychological first aid (PFA) was provided in a small number of institutions (9%). The most frequent symptom requiring psychological intervention in the PU group was anxiety, followed by insomnia and depression. In the PU group, supportive listening (75% of the 61 institutions that provided psychological intervention) and individual counseling (36%) accounted for most of the psychological interventions, and PFA was provided in some institutions (7%) (Figure 7).

The number of sessions and intervention duration varied among institutions. In the MH group, five institutions (23% of the 22 institutions that provided psychological interventions) provided only one session, and seven (32%) provided 10 sessions or more of psychological intervention lasting for 3 months or more. In the PU group, two institutions (only 3% of the 61 institutions that provided psychological intervention) provided only one session, 13 (21%) provided 2 to 4 sessions lasting for less than one month, 23 (38%) provided 5 to 9 sessions lasting for less than 3 months, and 16 (26%) provided 10 sessions or more lasting for 3 months or more. It appears that the number of sessions or intervention durations differed according to the severity of the situation (consultation for severe cases, e.g., suicidal ideation).

### 3.4. Remote Consultation

In the MH and PU groups, remote consultation was performed in 42 (79%) of 53 and 17 (20%) of 84 institutions, respectively. In the MH group, the categories of consulters were patients (74%), family members (69%), and local residents (79%, with the highest number of consultations). In the PU group, most consultations were by patients or family members. Of note, consultations were also made by staff in general and university hospitals, although the number was small. The most frequent methods of publication of remote consultation in the MH group were via the homepage (86%), by public relations magazines (52%), and posters (43%). The contents of consultation varied. The consultations made by the patients of the MH group were mostly about interpersonal problems (with family members, friends, and colleagues; 65%), mental symptoms (depressive symptoms; 55%), prejudice and discrimination (55%), anxiety and fear of being infected (50%), and anxiety and fear of infecting family members and colleagues (50%). The number of consultations according to the category of consulter in mental health and welfare centers are shown in Figure 8.

## 4. Discussion

To our knowledge, this is the first survey in Japan that examined the status of mental health care related to COVID-19, using a questionnaire survey of mental health and psychiatric institutions. The results showed that public mental health institutions, called “mental health and welfare centers,” are playing a central role in providing mental health consultation. All 55 mental health and welfare centers located in the prefectures and government-designated cities in Japan provide telephone consultation. The mean number of consultations per institution during the 10 months was 236, and the total number of consultations exceeded 12,000. On the other hand, the number of telephone consultations was small in psychiatric institutions (including psychiatric departments of general and university hospitals, private psychiatric hospitals, and psychiatric clinics); however, 90% or more of the psychiatric institutions that were consulted provided face-to-face consultations or examinations. One reason for this may be that consulters requiring treatment visited the medical institutions personally. Another reason for more face-to-face consultation via psychiatric institutions could be their relatively more serious cases.

The number of consultations increased from April 2020, when a national state of emergency was declared in Japan, and then the number of consultations gradually decreased along with the reduction in the number of infections. However, in Japan, there was a sharp increase in the number of COVID-19 positive cases from December 2020 (Figure 1). Given that the number of infected people was markedly higher compared to those in the first and second waves of the epidemic in Japan, there is a concern that the number of mental health consultations related to COVID-19 will rapidly increase.

The contents of consultation varied, including anxiety, depressive mood and loss of motivation, interpersonal and child problems, psychosomatic problems, and insomnia, suggesting a significant impact of COVID-19 on mental health. Almost all mental health and psychiatric institutions were consulted for psychological problems. In mental health and welfare centers, many of the consultations were provided by telephone, and almost all the centers were consulted for both new and worsening of pre-existing psychological problems. Approximately 30% of the institutions (both MH and PU) received consultations requiring urgent care, with a mean number of 4–5 consultations, indicating that a certain number of consultations required specialty psychiatric care, although the number was relatively small. In addition, there were consultations for serious problems such as suicidal ideation and attempt. In fact, the number of suicides in Japan increased from July to October 2020 [12]. According to a survey conducted at Tokyo University [13], the suicide rate in males increased between October and November while that in females increased from July to November, 2020, compared to these months from 2016−2019. In particular, the increases in suicide rates were most pronounced among men younger than 30 years and women aged below 30 years and 30–49 years. A research group at Kyoto University suggested that the annual number of suicides in Japan may increase up to 40,000 due to the influence of COVID-19 [10]; therefore, suicide prevention is an urgent issue. These findings suggest a need to establish a mental health triage system for mental health services during the COVID-19 pandemic based on the already established mental health triage system for disasters [14].

Psychological intervention was provided in approximately 40% and 70% of MH and PU institutions, respectively, that were consulted. Although the number of sessions and intervention duration varied among MH institutions, approximately 30% of interventions consisting of 10 sessions or more or lasting for 3 months or more occurred in MH institutions, suggesting that they spent much time and effort in practice. In psychiatric institutions, the most common methods of intervention were interviewing and counseling. In MH institutions, the methods of intervention varied, including health consultation, utilization of brochures, holding workshops, individual counseling, and referral to medical institutions.

Systematic psychotherapy, including cognitive behavior therapy (CBT), was not sufficiently provided in MH or PU institutions. The results of this questionnaire survey showed that the degree of mental health difficulties varied greatly. For patients with mild depression, international guidelines [15,16,17] recommend psychotherapeutic approaches such as CBT and problem-solving therapy rather than drug therapy. It is expected that symptoms are mild or below the threshold of mental disorders in most of the individuals consulting for mental health difficulties; therefore, it is desirable to be prepared for psychological intervention such as CBT. In Japan, CBT is not adequately reimbursed by public health insurance, for which an improvement is needed. It is also important to be prepared for web-based CBT or low-intensity CBT that is easy to learn.

Psychological first aid (PFA) is an approach of early intervention designed to help people during disasters. PFA is designed to reduce the initial distress caused by traumatic events and to develop adaptive functioning and coping. It was, however, used only in a small number of institutions in the present survey. This may partly be because pandemics are different from natural disasters, and conventional PFAs, such as the WHO [18] and the U.S. versions [19], were designed to help people in general. In the future, it may be useful to establish a PFA model tailored to address COVID-19, as a rapid PFA program that takes specialized intervention by psychiatric care workers into consideration [20].

Remote consultation was performed in approximately 80% of the MH institutions. For the purpose of suicide prevention, the need to develop remote psychological assessment and care and staff training programs in MH institutions has been highlighted worldwide [21]. Therefore, there is a need to explore the benefits of remote mental health services, especially for suicide prevention, in MH institutions in Japan. Consultations were made not only by COVID-19 patients and family members but also by local residents, who formed the most common category of consulters, suggesting that the prevalence of anxiety related to COVID-19 was high among the general population. The contents of consultation included anxiety and fear of being infected, as well as interpersonal problems (family, friends, colleagues) and prejudice and discrimination (reported by almost half of the MH institutions); this indicates that the impact of COVID-19 was wide-ranging, with a possibly large impact on our society. Furthermore, Imamura et al. [22] conducted a large-scale randomized controlled trial of internet-based CBT in workers and showed that it was effective in preventing major depression recurrence. Internet-based CBT could be a useful method during the COVID-19 pandemic because its effectiveness has been demonstrated in several studies [23,24].

This questionnaire survey had some methodology limitations. First, the response rate was relatively low, and thus, the study institutions were not representative of all institutions in Japan. Especially, the very low response rate in the PU group may suggest a biased sample of psychiatric institutions. Second, many consultations may have also been made in general medical and private institutions, but such data were unavailable. Third, from this study, it is unclear what percentage of individuals needed support and actually received consultation; of note, previous studies showed that less than half of psychiatric patients meeting the diagnostic criteria visited medical institutions [25]. Further follow-up studies are needed to evaluate the impact of COVID-19 on mental health and to develop effective countermeasures against it.

## 5. Conclusions

This study investigated the status of mental health care related to COVID-19 using a questionnaire survey of mental health and psychiatric institutions in Japan. Many consultations for mental difficulties were made by telephone in mental health institutions. The contents of consultation varied, including anxiety, depression, insomnia, and alcohol problems. Consultations were made by Covid-19–positive people, family members, and the general population. The contents of consultation included anxiety related to infection, mental health symptoms, and psychosocial problems such as interpersonal problems, prejudice, and discrimination. In addition, consultations requiring urgent care were reported in psychiatric and mental health institutions. Mental health and welfare centers in Japan play an important role in identifying mental health needs and providing interventions. Based on the results of this study, we suggest that it is urgent to develop a mental health triage system and a brief psychological intervention method that are feasible for use in mental health and welfare centers, to address mental health problems associated with COVID-19. Furthermore, given the number of remote consultations, more remote interventions should be needed.

## Figures and Tables

**Figure 1 ijerph-18-07318-f001:**
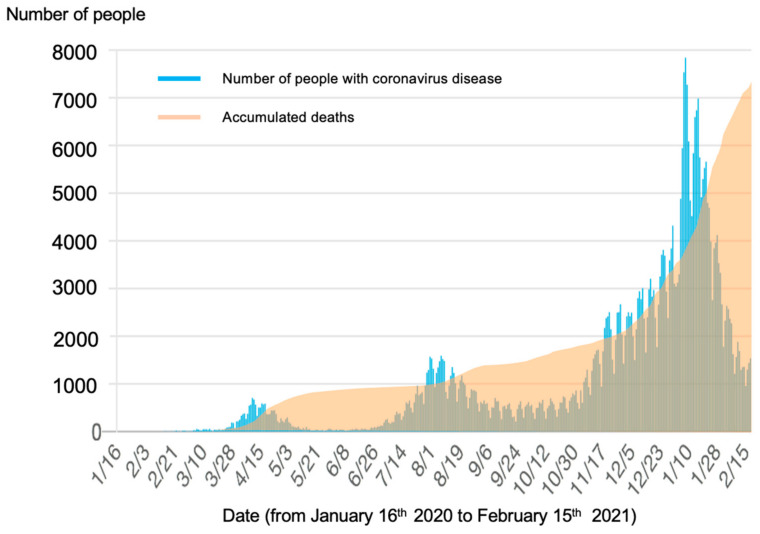
Changes in the number of people with coronavirus disease and accumulated deaths in Japan.

**Figure 2 ijerph-18-07318-f002:**
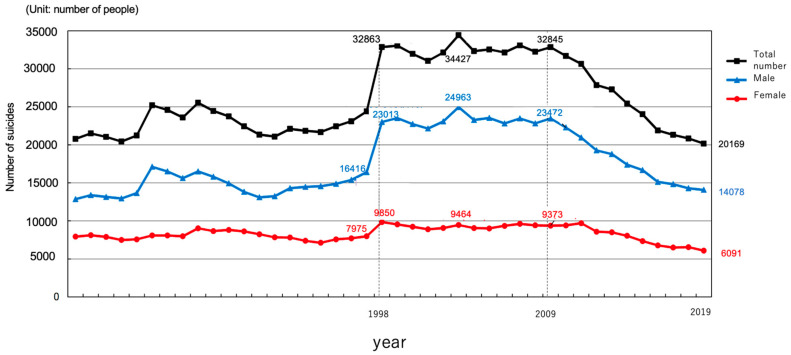
Annual change in the number of suicides in Japan from 1998 to 2019.

**Figure 3 ijerph-18-07318-f003:**
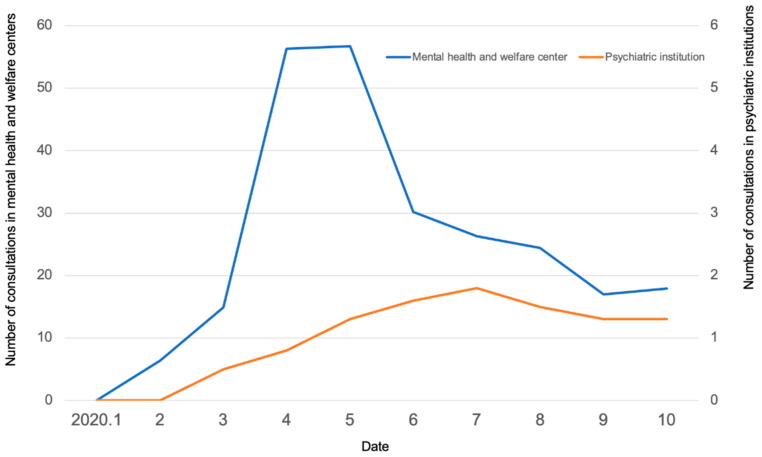
Number of consultations per month.

**Figure 4 ijerph-18-07318-f004:**
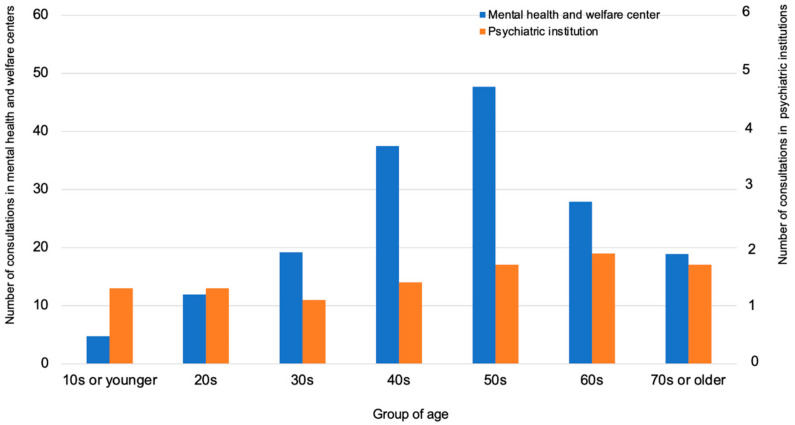
Number of consultations by age group.

**Figure 5 ijerph-18-07318-f005:**
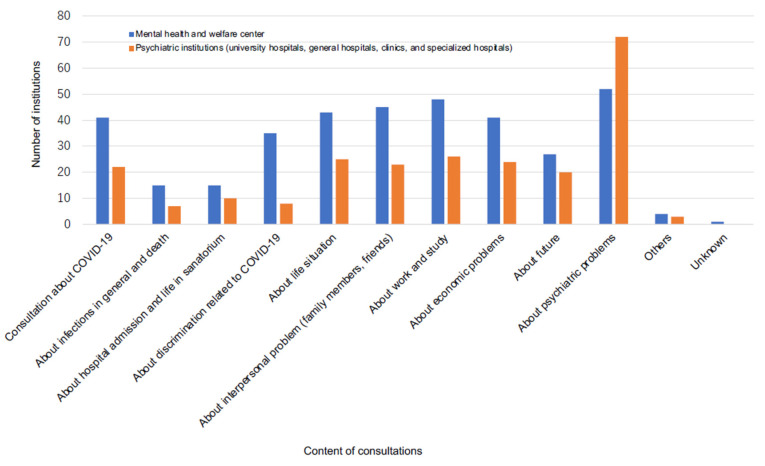
Number of consultations according to content.

**Figure 6 ijerph-18-07318-f006:**
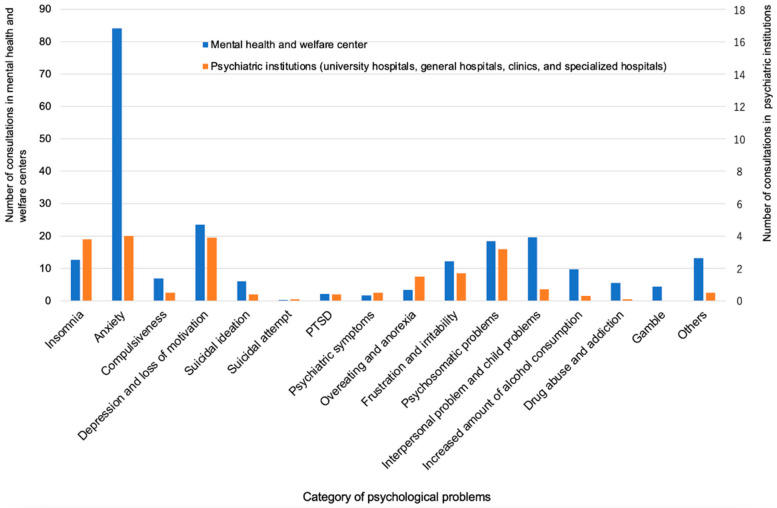
Categories of psychological problems.

**Figure 7 ijerph-18-07318-f007:**
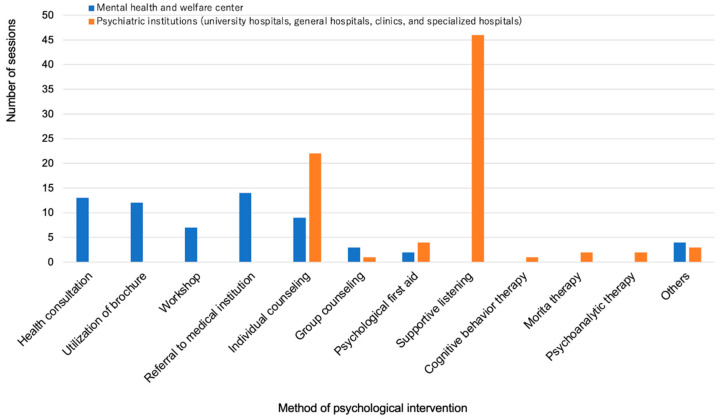
Number of sessions according to psychological intervention method.

**Figure 8 ijerph-18-07318-f008:**
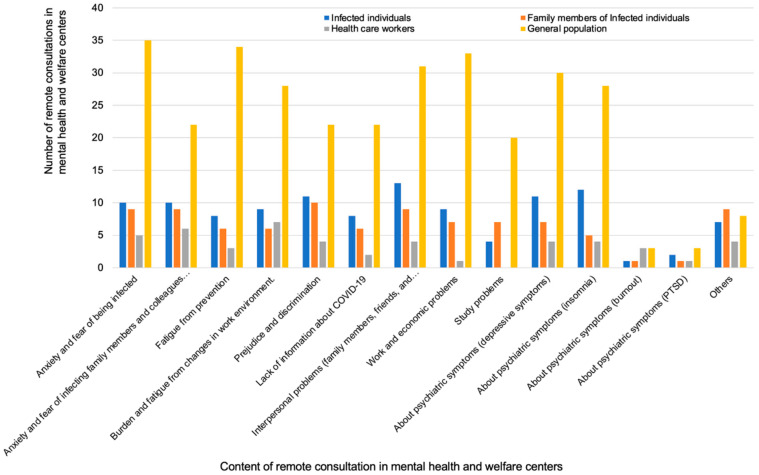
Number of remote consultations by content in mental health and welfare centers.

**Table 1 ijerph-18-07318-t001:** Outlines of the results of the questionnaire survey.

	Mental Health and Welfare Center	%	Psychiatric Institution	%
Number of institutions to which the questionnaire was sent	69		931	
Number of institutions that responded to the questionnaire	55	80	194	21
Psychiatric department of general hospitals/university hospitals			67	
Psychiatric clinic			84	
Psychiatric specialized hospital			43	
Provided consultation	53	96	84	43
Psychiatric department of general hospital/university hospital			33	
Psychiatric clinic			34	
Psychiatric specialized hospital			17	
Method of consultation (multiple answers allowed)				
Telephone	53	100	22	26
Email	4	7.5	4	4.8
Face-to-face consultation/examination	15	28	80	95
Number of consultations (total, from January to October, mean per institution)				
Telephone	236		9	
Email	2		2	
Face-to-face consultation/examination	11		7	
Number of consultations by sex (total, from January to October, mean per institution)				
Female	152		6	
Male	87		4	
Consultation requiring urgent care				
Yes	15	28	28	33
Number of consultations (total, from January to October, mean per institution)	5.6		4.3	
Psychological intervention				
Present	22	41	61	72
Number of sessions and duration of psychological intervention				
Once	5	23	2	3
2 to 4 times or lasting for less than 1 month	4	18	13	21
5 to 9 times or lasting for less than 3 months	1	4	23	38
10 times or more or lasting for 3 months or more	7	32	16	26
Unknown	5	23	7	12
Remote consultation				
Done	42	79	17	20

## Data Availability

The data presented in this study are available on request from the corresponding author. The data are not publicly available due to privacy protection.

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
