# Peer review of "Mental Health Difficulties and Countermeasures during the Coronavirus Disease Pandemic in Japan: A Nationwide Questionnaire Survey of Mental Health and Psychiatric Institutions"

_ijerph, 2021, doi:10.3390/ijerph18147318_

Round 1

Reviewer 1 Report

I think this paper is of value to readers based in Japan and outside to consider the changes and utilisation of mental health care in the context of the ongoing COVID-19 pandemic. Comments are below.

Introduction

Largely very well written, concise, and clear. Some small suggested edits to improve the introduction:

Line 66 – I would rephrase “mentally impaired patients” to something like “patients with mental illhealth” or “patients with a mental health disorder” to keep in line with people-first language and to stay away from the idea that a mental health illness is equivalent to an impairment from a “preventing stigma” point of view.

Line 69 – I am unsure what is meant by a “compulsive symptom” – this could be referring specifically to the compulsions seen in an OCD presentation or in more behavioural terms in other disorders. I would suggest making this clearer for the reader by defining or changing the wording.

Line 79 – I wonder whether presenting the suicide frequency as a rate in comparison to the size of the overall population of Japan would be helpful, as I can see from the numbers that there has been a raw decrease of 10,000 suicides (which is great), but unsure how that compares to the populations. I assume the population of Japan has increased as it has with many other countries, however if this is not the case and it has decreased, the decrease in suicides may be a by-product of a smaller population for example. Therefore, a rate would just help the reader understand this is not the case.

Line 89 – again, I would stay away from terms like “mentally challenged people” as it suggests there is some defect or deficit and that the person is entirely defined by their mental health difficulty. This term, for western cultures such as the UK, is also an outdated term to describe those with a learning/intellectual disability and thus may be confused as such by international readers in this international journal. Therefore, I would suggest an amendment in line with my suggestion above.

Materials and Methods

Line 106 and 107 – this briefly describes the Design study and thus I would suggest adding a subtitle to help the reader easily find the Design section should they wish to.

Robust sampling method.

The consent procedures seem a little unclear as to whether they were informed consent or simply consent by process of completing the questionnaire. It would be good to know whether the original information sheet/request form explained the key areas of informed consent (i.e. participation is voluntary, withdrawal was possible without consequence, confidentiality, etc) so the reader can judge whether the participants did indeed give informed consent (especially as this is now included in many methodological quality rating tools for research).

Line 134 – I would suggest changing “mental problem” to something like “mental health difficulty” or “disorder”, for the reasons detailed above about language. Similar suggestion for such wording hereafter.

Results

Note the very low response rate/high attrition rate may suggest a biased sample. It is likely of interest to the reader that the response rate was higher in mental health and welfare centres compared to the other settings included – it will be important to make sure this is covered in the discussion and limitations section.

Figures – general note to make sure all axes in your figures are labelled so they are clear to the reader. Otherwise, I appreciate the use of figures to aid interpretation of the findings.

Line 226 – I am unsure how “abuse” is a symptom of a mental health difficulty? Rather it would be seen as an antecedent or predisposing factor perhaps? I would suggest clarifying this wording as I assume the psychological intervention would be for the psychological impact of any potential abuse rather than treating the abuse itself?

Discussion

I would suggest being a little more tentative with your conclusions regarding psychiatric institutions due to the high subject attrition from this sample, meaning it may be that results cannot generalise and that there may be biases in your sample.

General note throughout that when referring to “infected people” it would be good, if it is not mentioned in a close by sentence, to clarify that this means people that were COVID positive or infected with COVID-19.

Reviewer 2 Report

Thank you for the opportunity to review this article, which surveyed mental health and psychiatric institutions across Japan to describe patterns in mental health service uptake during the 2020 year due to Covid-19.  I have the following suggestions for improving the clarity and validity of the manuscript.

  1. Reword abstract lines 29-30 so that the final sample size is clearer. The 69 and 931 were the number of questionnaires sent, whereas the 55 and 194 were the final analysed surveys. This is not immediately clear.  Also, are you sure you can call this a "nationwide survey" in your title given the low response rate, and your note in the discussion that results may not be generalisable to all of Japan?
  2. Abstract - more detail is needed about what was included in the survey.
  3. Line 44 needs grammatical editing. 
  4. Line 57-58- is there no counter evidence that mental health problems decreased during the Covid-19 pandemic?
  5. Section 2.1: What percentage of total institutions across Japan were randomly selected to be invited? Did you only invite a subset of psychiatric hospitals? You need to make it easier to gauge the Total # of institutions,  % invited, % responded. 
  6. Section 2.2 - be clearer that the expectation was for one survey to be completed for each institution. Why are lines 118 - 123 seemingly duplicated - how are those convening different information?  Also, how did administrators complete the survey-- were they asked to look up the data in their records or did they estimate it by memory? What instructions and controls were there over how they completed the survey. 
  7. Line 144 Change "Information were" to "Information was".
  8. Section 3.1 suggest rewording: "Responses were obtained from a total of 249 of 1,000 institutions surveyed (response rate, 25%), consisting of 55 mental health....". Also, is there any evidence that your response rates were tied to geographic location or population density or other factors that could affect generalisability?
  9. Line 170 - why is this timeframe Jan - Oct, 2020 different from the Timeframe you stated in the methods (Jan - Sept, 2020)?
  10. Figure 3 - for this figure you need to state what the two axes are labeled. What does the 0 to 60 refer to and what does the 0 to 6 refer to? 
  11. Line 270 - You state "All 69 mental health and welfare centers"... do you mean all 55 centers that were surveyed? You did not gather data on all 69 - unless you know this information aside from your survey. 
  12. Line 277- another reason for more face-to-face consultation via psychiatric institutions could be their relatively more serious cases. 
  13. Line 326- define PFA
  14. Line 369 - Another implication of your findings is the need for more remote interventions (tele health or eHealth or mHealth), given the number of remote consultations. 

Round 2

Reviewer 2 Report

The authors have addressed all of my suggestions - thank you.